Traumatic spinal cord injury: identifying independent risk factors and predictive model development for symptomatic urinary tract infections

Du Huayong 1
Li Zehui 1
Zhang Jinming 1
Wang Xiaoxin 1
Jing Yingli 1 2
Yang Degang 17610858918@163.com 1 3
Li Jianjun 2246476932@qq.com 1 3
1 School of Rehabilitation Medicine, Capital Medical University , Beijing , Fengtai , China
2 China Rehabilitation Science Institute , Beijing , Fengtai , China
3 Department of Spinal and Neural Functional Reconstruction, China Rehabilitation Research Center , Beijing , Fengtai , China
Marunaka Yoshinori
Electronic publication date: 2025 May 28
Publication date: 2025
Volume: 13
Electronic Location ID: e19473
Received 2024 Dec 1; Accepted 2025 Apr 24
Copyright: ©2025 Du et al.
Copyright year: 2025
Copyright holder: Du et al.
License: This is an open access article distributed under the terms of the Creative Commons Attribution License, which permits unrestricted use, distribution, reproduction and adaptation in any medium and for any purpose provided that it is properly attributed. For attribution, the original author(s), title, publication source (PeerJ) and either DOI or URL of the article must be cited.
License URL: https://creativecommons.org/licenses/by/4.0/

Keywords: Traumatic spinal cord injury, Symptomatic urinary tract infections, Risk factors, Logistic regression, Receiver operator characteristic

Funding: The Beijing Municipal Science and Technology Commission’s Capital Clinical Characteristic Diagnosis and Treatment Technology Research and Application Z221100007422044 The Key Project of China Rehabilitation Research Center 2021ZX-07 This study was funded by the Beijing Municipal Science and Technology Commission’s Capital Clinical Characteristic Diagnosis and Treatment Technology Research and Application (Z221100007422044). The publication fee was funded by the Key Project of China Rehabilitation Research Center (2021ZX-07). The funders had no role in study design, data collection and analysis, decision to publish, or preparation of the manuscript.

==============================
Background

Traumatic spinal cord injury (TSCI) is commonly associated with urinary tract infections (UTIs), with a reported prevalence ranging from 31.7% to 68%. Symptomatic UTIs can result in serious complications, including chronic kidney damage and recurrent infections. The objective of this study was to identify independent risk factors and develop a predictive model for symptomatic UTIs in TSCI patients, thereby providing valuable insights for prevention and management strategies.

Methods

A retrospective study was conducted at the China Rehabilitation Research Center, involving 168 TSCI patients admitted between January 1, 2020, and August 1, 2024. Symptomatic UTIs were diagnosed using Delphi consensus criteria, which integrated clinical symptoms, urinalysis, and culture confirmation. Comprehensive clinical data, including demographic characteristics, injury profiles, and laboratory parameters, were systematically extracted from the hospital information system. Potential risk factors were initially screened using univariable logistic regression, with statistically significant variables subsequently analyzed in a multivariable logistic regression model to identify independent predictors. A predictive model for symptomatic UTIs was constructed using the regression coefficients. The model’s performance was evaluated using the area under the receiver operating characteristic curve (AUC), calibration with the Hosmer-Lemeshow test, and internal validation through bootstrap resampling.

Results

The incidence of symptomatic UTIs was 57.14%, with the majority presenting with fever (65.07%) and Escherichia coli infections (44.52%). Prolonged hospitalization (OR = 1.005, 95% CI [1.001–1.010]) and cumulative antibiotic exposure (OR = 1.011, 95% CI [1.000–1.022]) were identified as independent risk factors. The predictive model, which incorporated these factors, demonstrated strong discrimination (AUC = 0.81, 95% CI [0.746–0.879]) and good calibration (P = 0.44).

Conclusions

This study presents the incidence of symptomatic UTIs in TSCI patients and identifies two critical predictive factors along with a risk score for early prediction of symptomatic UTIs. The findings provide a foundation for improved clinical practices aimed at preventing and managing symptomatic UTIs in this patient population, potentially reducing healthcare costs and improving patient outcomes.

Introduction

Traumatic spinal cord injury (TSCI) is a severe central nervous system disorder commonly associated with complications (Ahuja et al., 2017), with urinary tract infection (UTIs) being the most frequent, ranges from 31.7% to 68% (Esclarín De Ruz, García Leoni & Herruzo Cabrera, 2000; Kim et al., 2021). UTIs are classified as symptomatic or asymptomatic based on the presence of clinical symptoms (Hooton, 2012). Symptomatic UTIs present with frequent urination, urgency, dysuria, lower abdominal pain, and fever. After ruling out other infections through urinalysis and urine culture, a diagnosis of symptomatic UTIs is confirmed. UTIs not only increase healthcare costs and hospital stays but also cause chronic kidney damage, recurrent infections, and life-threatening complications (Bavanandan & Keita, 2024).

The high incidence of symptomatic UTIs can be attributed to several factors, the most direct of which is nerve damage, leading to bladder dysfunction, urinary retention, and a conducive environment for bacterial growth (D’Hondt & Everaert, 2011). Nerve repair is a protracted process, and currently, no established technology exists to reconnect damaged spinal cord tissue (Stokes, Drozda & Lee, 2022). Consequently, effective management of symptomatic UTIs relies primarily on prevention strategies and careful management. In clinical practice, preventive measures include maintaining urinary tract patency, regular catheterization, and improving hydration (Musco et al., 2022). Once symptoms emerge, antibiotics are selected based on urine culture results (Tandan et al., 2016). However, addressing the underlying causes requires further research and a better understanding of UTI risk factors. Although several studies have explored UTI risk factors in TSCI patients, most have focused on individual factors and have not thoroughly examined their combined effects. Furthermore, many studies have small sample sizes, and the time since injury varies considerably among patients. There is also a lack of focused analysis on the risk factors specific to symptomatic UTIs, which may be affected by inconsistent diagnostic criteria (Goodes et al., 2020; Kim et al., 2021; Liu et al., 2024).

This study analyzed the clinical data of TSCI patients admitted within one month of injury. The primary objective was to identify independent risk factors and develop a predictive model for the early detection of symptomatic UTIs. The findings aim to inform preventive strategies, optimize management practices, and ultimately enhance patient outcomes.

Materials & Methods

Study design and participants

The China Rehabilitation Research Center (CRRC) is China’s first Class-A tertiary public hospital specializing in rehabilitation. It comprises three primary departments focused on treating spinal cord injury: the Department of Spinal Cord Injury Rehabilitation (A), the Department of Spinal Cord and Nerve Function Reconstruction (B), and the Department of Spinal Cord Surgery (C). Each year, CRRC serves over 500 spinal cord injury patients from across the country. This study is retrospective and is part of the Capital Clinical Characteristic Diagnosis and Treatment Technology Research and Transformation Application Project, received ethical approval from the China Rehabilitation Research Center Ethics Committee (Approval No. 2022-143-01) and adheres to the Declaration of Helsinki. Upon the commencement of the project on November 25, 2022, written informed consent was secured from all subsequently enrolled participants. For retrospective data collected before this date, 93 eligible patients were identified through the hospital information system. These individuals were contacted via telephone to obtain verbal consent for the use of their data.

The inclusion criteria were as follows: patients diagnosed with TSCI (Ahuja et al., 2017), aged between 18 and 65 years, with no history of immune system diseases (e.g., ankylosing spondylitis, systemic lupus erythematosus, malignancies) or infectious diseases (e.g., tuberculosis, hepatitis), and with complete AISA examinations and blood tests upon admission. Exclusion criteria included non-traumatic spinal cord injuries, age outside the 18–65 range, a history of immune or infectious diseases, incomplete AISA examinations or blood tests, and pregnancy. A total of 168 TSCI patients met the criteria and were included in the study.

Clinical data

Data from each patient’s file and medical records were anonymized to remove any identifiable information. All methods adhered to relevant guidelines and regulations. A self-designed standardized form was used to collect basic demographic and clinical information for all patients who met the inclusion criteria. The recorded data included age, gender, medical history, causes of injury, creatinine and albumin level at admission, length of hospital stay. The times of antibiotic use and indwelling catheters use during hospitalization can be obtained through the hospital infection system. Proteinuria is assessed through routine urinalysis, with the following classifications: “−” denotes the absence of proteinuria, “+” indicates mild proteinuria, “++” represents moderate proteinuria, and “+++” reflects severe proteinuria. According to the Polytrauma-Schlüssel scoring system (Teijink et al., 1993), TSCI combined with at least one injury to another body region was classified as polytrauma. The level and severity of spinal cord injury were assessed based on the admission physical examination, using the International Spinal Cord Society’s online tool (https://isncscialgorithm.com/) and American spinal injury association impairment scale (AIS) (Rupp et al., 2021).

Definition of symptomatic UTIs

A recently published multidisciplinary Delphi consensus in The Lancet provided diagnostic procedures and criteria for symptomatic UTIs (Bilsen et al., 2024). The consensus divided the diagnosis into four major modules: “local symptoms and signs,” “systemic criteria,” “urinalysis,” and “urine culture,” with 15 specific indicators in total. Scores were assigned to each module (0–3 points). A total score of ≥ 8 points confirmed symptomatic UTIs, significant bacteriuria (≥ 50 white blood cells/hpf), a positive urine culture, and at least one local symptom or sign (e.g., dysuria, urgency, frequency, mild suprapubic pain or tenderness, flank pain or tenderness, perineal pain, prostate pain, incontinence, or gross hematuria) and/or systemic criteria (e.g., fever, C-reactive protein ≥ 50 mg/L, procalcitonin ≥ 0.5 ng/mL, or bacteremia). Symptomatic UTIs was diagnosed independently by two experienced physicians based on clinical and laboratory data, with any discrepancies resolved through discussion with a third physician.

Establishment and validation of the prediction model

Patients were stratified into a symptomatic UTI group and a no symptomatic UTI group according to the presence or absence of clinical symptoms indicative of urinary tract infection. The no symptomatic UTI group encompassed both individuals with no evidence of urinary tract infection and those diagnosed with asymptomatic bacteriuria. To identify risk factors and independent predictors, variables with P-values < 0.05 in the baseline analysis were included in univariable logistic regression. Those demonstrating P-values < 0.05 in the univariable analysis were further analyzed using multivariable logistic regression. Finally, variables without collinearity and exhibiting P-values < 0.05 in regression analysis were selected for model construction. The risk score was computed using the following formula: risk score = (factor 1 × α) + (factor 2 ×β) + ... + (factor n  ×γ), where α, β, and γ represent the regression coefficients.

Statistical methods

Statistical analyses were carried out using SPSS 27.0.1. Continuous variables adhering to the assumptions of normal distribution and homogeneity of variance were summarized as mean ± standard deviation, with group comparisons conducted via independent samples t-tests. For variables deviating from these assumptions, medians and interquartile ranges were reported, and non-parametric Mann–Whitney U tests were utilized for intergroup comparisons. Categorical variables were presented as frequencies and percentages, with chi-square tests employed for group comparisons. Both univariate and multivariate regression analyses were performed, with statistical significance defined as P < 0.05.

For more advanced analyses, Python 4.4.2 was implemented. Categorical variables were assigned numerical codes as follows: symptomatic urinary tract infection (no = 0, yes = 1), injury level (cervical = 1, thoracic = 2), polytrauma (no = 0, yes = 1), AIS grade (AIS A = 1, AIS B = 2, AIS C = 3, AIS D = 4) and admitting department (1 = Department A, 2 = Department B, 3 = Department C). The analysis incorporated several key R packages, including car, pheatmap, pROC, ResourceSelection, and boot. Multicollinearity was examined using the variance inflation factor (VIF), with a threshold of VIF ≥ 10 indicating significant multicollinearity. The model’s discriminative capacity was evaluated through receiver operating characteristic (ROC) curves, with the area under the curve (AUC) computed. Model calibration was assessed using the Hosmer-Lemeshow test. Internal validation was conducted via bootstrap resampling with 1,000 iterations, and both discrimination and calibration metrics were documented.

Results

Descriptive statistics of clinical data for patients with TSCI

A total of 168 TSCI patients were included (Fig. 1), and their demographic and clinical characteristics are presented in Table 1. The average age of the patients was 42.6 0 years, with males comprising 84.52% of the total. The leading causes of injury were falls from heights (36.90%) and traffic accidents (25.60%). The average hospital stay was 168.90 days, and 57.14% of the patients developed symptomatic UTIs, with the highest frequency observed during the intermediate phase. Injuries were primarily located in the cervical and thoracic regions, with 46.43% of patients classified as AIS A, representing the highest proportion.

Figure 1 The flowchart of patients’ enrollment and grouping.

Table 1 Clinical characteristics of enrolled patients.

Projects	Data	
Sample size	168	
Age, x ± s	42.60 ± 12.36	
Gender, n (%)		
Male	142 (84.52)	
Female	26 (15.48)	
Cause of Injury, n (%)		
Traffic accidents	43 (25.60)	
Falls from heights	62 (36.90)	
Falls	35 (20.83)	
Crushed by heavy objects	25 (14.88)	
Others	3 (1.79)	
Length of Hospital Stay, x ± s	168.90 ± 134.48	
History of diabetes, n (%)	10.12% (17)	
Polytrauma, n (%)	35.12% (59)	
Symptomatic UTIs		
Proportion, n (%)	57.14 (96)	
Total person-times	155	
Disease progression, f (average person-times)		
Subacute phase	3 (0.07)	
Intermediate phase	137 (0.82)	
Chronic phase	15 (0.38)	
Injury level, n (%)		
Cervical	78 (46.43)	
Thoracic	77 (45.83)	
Lumbar	13 (7.74)	
AIS, n (%)		
AIS A	78 (46.43)	
AIS B	39 (23.21)	
AIS C	30 (17.86)	
AIS D	21 (12.5)	

Table 2 provides detailed information on the symptoms, urine culture, urinalysis, and blood test results of patients with symptomatic UTIs. Fever was the most common symptom, occurring in 65.07% of cases, while Escherichia coli was the most frequent pathogen, accounting for 44.52% of infections. All symptomatic UTIs patients had elevated leukocyte counts in urine tests, and 80% showed increased C-reactive protein (CRP) levels (≥ 50 mg/L), indicating the severity of the infection.

Table 2 UTIs symptoms and key laboratory tests of enrolled patients.

Symptoms	Frequency (%)	Urinalysis	Frequency (%)	
Fever	136 (65.07)	Escherichia coli	69 (44.52)	
Frequent urination/Urgency/Dysuria	15 (7.18)	Klebsiella pneumoniae	25 (16.13)	
Hematuria	14 (6.70)	Others*	12 (7.74)	
Incontinence	13 (6.22)	Proteus mirabilis	11 (7.10)	
Flocculent, foul-smelling, cloudy urine	12 (5.74)	Acinetobacter baumannii	7 (4.52)	
Worsening spasticity in lower limbs	9 (4.31)	Coagulase-negative staphylococci	4 (2.58)	
Worsening neuropathic pain in lower limbs	5 (2.39)	Enterobacter cloacae	3 (1.94)	
Discomfort in the groin area	5 (2.39)	Pseudomonas aeruginosa	3 (1.94)	
		Enterococcus faecalis	3 (1.94)	
		Enterococcus faecalis	3 (1.94)	
		Serratia marcescens	2 (1.29)	
		Gram-negative bacilli	2 (1.29)	
		Klebsiella oxytoca	2 (1.29)	
		Corynebacterium spp.	2 (1.29)	
		Stenotrophomonas maltophilia	1 (0.65)	
		Candida tropicalis	1 (0.65)	
		Morganella morganii	1 (0.65)	
		Gram-positive bacilli	1 (0.65)	
		Citrobacter freundii	1(0.65)	
		Enterobacter aerogenes	1 (0.65)	
		Candida albicans	1 (0.65)	

Risk factors analysis of symptomatic UTIs

A comparison of characteristics between symptomatic UTIs group (n = 96) and no symptomatic UTIs group (n = 72) is presented in Table 3. The incidence of symptomatic UTIs was significantly associated with times of antibiotic uses, times of indwelling catheter use, length of hospital stay, admitting department, polytrauma, injury level, and AIS grade (P < 0.05). Age, gender, albumin level and creatinine level did not show statistically significant differences (P < 0.05).

Table 3 Baseline analysis between symptomatic UTIs group and no symptomatic UTIs group.

Variable	No symptomatic UTIs (n = 72)	Symptomatic UTIs (n = 96)	Statistical value	P-value	
Age, Median (Q1, Q3)	41 (32.25, 53.75)	43 (33, 54)	−0.394	0.69	
Albumin (x ± s, g/L)	37.58 ± 3.02	36.70 ± 3.35	1.757	0.081	
Creatinine (x ± s, umol/L)	55.69 ± 13.40	58.61 ± 12.99	−1.416	0.16	
Times of antibiotic use, Median (Q1, Q3)	15 (3.25, 40.75)	60.50 (30.25, 116.75)	−6.545	<0.001	
Times of indwelling catheter use, Median (Q1, Q3)	24 (10, 46.75)	45 (26.25, 88.50)	−4.358	<0.001	
Length of stay, Median (Q1, Q3)	91.5 (52.25, 153.75)	162.5 (104, 328.25)	−4.802	<0.001	
Proteinuria, n (%)			2.480	0.48	
Negative	58 (80.56)	81 (84.37)			
+ Positive	10 (13.88)	11 (11.46)			
++ Positive	3 (4.17)	3 (3.13)			
+++ Positive	1 (1.39)	1 (1.04)			
Admitting department, n (%)			6.679	0.04	
Department A	40 (55.60)	64 (66.70)			
Department B	17 (23.60)	25 (26.00)			
Department C	15 (20.80)	7 (7.30)			
Gender, n (%)			1.517	0.218	
Male	58 (80.60)	84 (87.50)			
Female	14 (19.40)	12 (12.50)			
Polytrauma, n (%)			6.300	0.01	
No	54 (75.00)	54 (56.20)			
Yes	18 (25.00)	42 (43.80)			
Injury level, n (%)			10.592	0.005	
Cervical	33 (45.80)	45 (46.90)			
Thoracic	28 (38.90)	49 (51.00)			
Lumbar	11 (15.30)	2 (2.10)			
AIS grade, n (%)			9.939	0.02	
AIS A	28 (38.90)	50 (52.10)			
AIS B	14 (19.40)	25 (26.00)			
AIS C	15 (20.80)	15 (15.60)			
AIS D	15 (20.80)	6 (6.20)			
Notes.

Bold indicates significant findings.

Variables demonstrating statistically significant differences in the baseline analysis were incorporated into the regression models. In the univariate logistic regression analysis, several variables showed significant associations: times of antibiotic use (OR = 1.019, 95% CI [1.011–1.029], P < 0.001), times of indwelling catheter use (OR = 1.020, 95% CI [1.010–1.030], P < 0.001), length of hospital stay (OR = 1.008, 95% CI [1.004–1.012], P < 0.001), polytrauma (OR = 2.333, 95% CI [1.196–4.554], P = 0.013), admitting department (OR = 0.617, 95% CI [0.400–0.952], P = 0.029), and AIS grade (OR = 0.626, 95% CI [0.488–0.880], P = 0.005). In the multivariate logistic regression model, only times of antibiotic use (OR = 1.011, 95% CI [1.000–1.022], P = 0.04) and length of hospital stay (OR = 1.005, 95% CI [1.001–1.010], P = 0.009) retained statistical significance (Table 4).

Table 4 Regression analyses between symptomatic UTIs group and no symptomatic UTIs group.

	Univariable logistic analysis	Multivariable logistic analysis	
Variable	Regression coefficient	OR	95% CI	P-value	Partial regression coefficient	OR	95% CI	P-value	
Times of antibiotic use	0.019	1.020	1.011–1.029	<0.001	0.011	1.011	1.000–1.022	0.04	
Times of indwelling catheter use	0.020	1.020	1.010–1.030	<0.001	0.006	1.006	0.993–1.019	0.38	
Length of hospital stay	0.008	1.008	1.004–1.012	<0.001	0.005	1.005	1.001–1.010	0.009	
Admitting department	−0.482	0.617	0.400–0.952	0.03	−0.377	0.686	0.411–1.131	0.14	
Polytrauma	0.847	2.333	1.196–4.554	0.01	0.601	1.825	0.848–3.995	0.13	
Injury level	−0.364	0.695	0.425–1.136	0.15			
AIS grade	−0.422	0.626	0.488–0.880	0.005	−0.108	0.897	0.628–1.282	0.55	
Notes.

Bold indicates significant findings.

Pearson and Spearman correlation analyses were used to assess the relationships between variables (Fig. 2). Notable positive correlations include times of antibiotic use with times of indwelling catheter use (r = 0.50), times of antibiotic use with length of hospital stay (r = 0.46), and length of hospital stay with times of indwelling catheter use (r = 0.39). AIS grade is negatively correlated with times of indwelling catheter use (r = min0.33).

Figure 2 Correlation matrix of variables and correlation coefficient distribution.

ROC curves and AUC analysis

The multicollinearity analysis revealed that all variance inflation factors (VIFs) were below 10, indicating no concerns with multicollinearity. The VIFs were as follows: times of indwelling catheter use (1.55), admitting department (1.05), times of antibiotic use (1.39), length of hospital stay (1.11), polytrauma (1.13), injury level (1.49), and AIS grade (1.55). As illustrated in Fig. 3, the AUC of multivariate regression model was 0.81 (95% CI [0.746–0.879]), indicating a significant capability in predicting the incidence of symptomatic UTIs. The AUC of univariate regression model was higher (AUC=0.93, 95% CI [0.746–0.879]). The Hosmer-Lemeshow test yielded a chi-square value of 7.972 with a p-value of 0.44, indicating that the model fits the data well. Based on Bootstrap validation, the model demonstrated a discriminative ability of 0.81 (95% CI [0.736–0.864]) and a calibration metric of 0.0157 (95% CI [0.002–0.0312]), seen in Appendix 1.

Figure 3 ROC analysis and curves drawing of predictive model for symptomatic UTIs.

Discussion

Independent risk factors for symptomatic UTIs

Times of antibiotic use and length of hospital stay demonstrated statistically significant associations in both univariate and multivariate regression analyses. Although antibiotics are commonly used for treating and preventing infections (e.g., weekly use of ciprofloxacin or nitrofurantoin) and have been shown to reduce the incidence of symptomatic UTIs (Jent et al., 2022), this study found that each additional use of antibiotics increased the risk of symptomatic UTIs by approximately 1.1% (OR = 1.011, P = 0.011), which may be related to increased antibiotic resistance and microbial imbalance (Jernigan et al., 2020; Luchen et al., 2023). These findings highlight the critical need for implementing antibiotic stewardship programs for TSCI patients. This includes restricting prophylactic antibiotics to high-risk subgroups, such as those with a history of recurrent UTIs, prioritizing culture-guided narrow-spectrum agents and closely monitoring microbiome restoration after antibiotic therapy. Patients who experienced symptomatic UTIs had longer hospital stay, likely due to multiple factors such as antibiotic use, hospital-acquired cross-infections, and frequent invasive urinary procedures. Correlation analysis (R = 0.46, P < 0.05) supports this observation. Therefore, effective prevention and management of symptomatic UTIs in TSCI patients should include early diagnosis and screening, minimizing the use of indwelling catheters, optimizing bladder management, boosting immunity, and providing health education.

Risk factors for symptomatic UTIs

Times of indwelling catheterization, admitting department, polytrauma, and AIS grade exhibited statistically significant associations in the univariate regression analysis. The use of an indwelling urinary catheter and the duration of catheterization are key contributors to the risk of catheter-associated urinary tract infection. The CDC recommends using catheters only when clinically indicated and switching to intermittent catheterization when no longer needed (Gould et al., 2010). Even intermittent catheterization is associated with symptomatic UTIs but at a lower rate than long-term indwelling catheters (Vahr et al., 2013). Violent events not only cause spinal cord injuries but may also result in injuries to the chest, abdomen, pelvis, or limbs. The most common accompanying injuries in TSCI patients are limb fractures, rib and scapular fractures (Wang et al., 2020), followed by lung contusions and brain injuries. This study found that TSCI patients with polytrauma had a higher incidence of symptomatic UTIs. The highest incidence of symptomatic UTIs was in Department A, while Department C had the lowest. Possible reasons include differences in cross-infection risks (Pittet et al., 2008), with Department A being close to the rehabilitation department and having more contact with hospitalized patients across the institution, thus increasing the risk of cross-infection. Moreover, the frequency of antibiotic use varied between departments (Kasse et al., 2024), with Department A having the highest usage (78.37 ± 104.27) and Department B the lowest (36.48 ± 52.55), a statistically significant difference. Additionally, variations in medical record documentation and symptomatic UTIs diagnostic criteria between departments could directly affect the reporting of symptomatic UTIs. For example, Department C had the lowest incidence of symptomatic UTIs, but some suspected symptomatic UTIs cases were often managed based on blood and urine tests combined with clinical experience, without conducting urine cultures, and thus were not included in the analysis. This variability emphasizes the need for standardized protocols across departments. Key measures include centralized training on aseptic catheterization techniques (Gould et al., 2010), thorough environmental decontamination of high-touch surfaces in rehabilitation areas, and mandatory detailed documentation of UTI diagnostic criteria. The higher the level of spinal cord injury (e.g., cervical or thoracic) and the more severe the injury (e.g., AIS A or AIS B), the greater the impairment of autonomic nervous system and bladder control functions. This leads to longer bed-ridden and more frequent catheter use, thereby increasing the risk of symptomatic UTIs (Winkelman, 2009). Baseline analysis, univariate regression, and previous studies support this conclusion (Kim et al., 2021; Moon et al., 2021), though some differences were noted in multivariate analysis, possibly due to confounding factors and collinearity issues.

Potential risk factors for symptomatic UTIs

This study did not find significant correlations between gender, age, or albumin levels and symptomatic UTIs. The study included only 26 female TSCI patients, limiting the assessment of gender’s influence on symptomatic UTIs and other complications. Most studies have reached similar conclusions, though Liu Jiawei et al. suggested that gender might be a factor influencing symptomatic UTIs. However, male TSCI patients are more likely to have indwelling catheters, so symptomatic UTIs occurrence may be more related to catheter use than gender itself. Catheterization, as an important medical procedure, is closely associated with urinary tract infections. The longer an indwelling catheter is used, the higher the infection risk, typically increasing by 3–7% per day (Patel et al., 2023). This study also showed a significant difference in infection risk between short-term (less than 30 days) and long-term (more than 30 days) indwelling catheter use. Albumin, a plasma protein synthesized by the liver, reflects the patient’s nutritional status to some extent (Thalacker Mercer & Campbell, 2008). Li et al. (2023)’s meta-analysis indicated that hypoalbuminemia is an important risk factor for urinary tract infections after spinal surgery, which aligns with Liu Jiawei’s findings. This may be because low albumin levels weaken the immune system, increasing susceptibility to infection (Calder, 2021). In this study, TSCI patients with symptomatic UTIs had lower albumin levels than those without symptomatic infection, with a P-value approaching 0.05.

Prediction model for urinary tract infection after spinal cord injury

This study used a stepwise statistical approach to develop a predictive model, drawing on methodologies from previous literature (Cheng et al., 2024; Huque et al., 2024). By integrating a risk stratification scheme, we enhanced the clinical applicability of the multivariate regression model (Risk score = 0.011  × antibiotic use + 0.005  × hospital stay). Each coefficient reflects the contribution of a variable to the risk score, with positive values indicating increased risk and negative values indicating reduced risk. The duration of antibiotic use carries more weight (0.011 vs. 0.005), emphasizing the need for careful antimicrobial management. While the univariate model yielded a high AUC (0.93), this likely reflects an “artificial predictive advantage” due to the omission of variable interactions. In contrast, the multivariate model, which produced a slightly lower AUC (0.81), takes the relationships between variables into account. With its better fit and robustness, the multivariate model is more suitable for real-world applications. The improved model now offers practical thresholds for stratified management. High-risk patients require closer monitoring (daily urine tests) and early guidance on antibiotic reduction based on culture results, while low-risk patients benefit from interventions like shorter catheterization time and hydration protocols. Additionally, the coefficient for hospital stay (0.005) supports clinical strategies for optimizing rehabilitation pathways and implementing telemedicine follow-up to guide discharge planning.

Comparison with previous studies

The incidence of symptomatic UTIs in this study was 57.14%, similar to the results of Liu et al. (2024) in China, but significantly higher than the 31.7% reported by Kim et al. (2021) and lower than the 71.8% reported by Milicevic et al. (2024). This variation may be due to differences in the diagnostic criteria for symptomatic UTIs, regional healthcare conditions, and the management of SCI patients. Moreover, this study found that patients in the intermediate phase of TSCI had the highest incidence and frequency of symptomatic UTIs, with fever being the most commonly recorded symptom, and Escherichia coli being the most common pathogen. This is consistent with findings from both domestic and international studies (Fitzpatrick & Nwafo, 2024; Milicevic et al., 2024), underscoring the importance of this pathogen in spinal cord injury patients. Future research should clarify symptomatic UTIs prevalence and dominant pathogens through multi-center studies, considering regional differences, to develop more targeted prevention strategies.

This study developed a predictive model that integrates clinical variables and laboratory indicators, demonstrating comparable efficacy to the findings reported by Zhao et al. (2024). Notably, they also identified white blood cell count as an independent risk factor for UTIs in hospitalized SCI patients. Compared to the predictive model developed by Si et al. (2019), which was based on scale assessments and systemic inflammatory response syndrome criteria, our model exhibits significant advantages, particularly given the inherent limitations of scale-based assessments in achieving complete objectivity. Recent research has increasingly focused on the development of machine learning models to predict infection risk across various healthcare settings. These models integrate multidimensional clinical data to provide early warnings of infection prior to clinical suspicion (Feng et al., 2023; Hassan et al., 2023). While these models demonstrate excellent theoretical predictive performance, with AUC values often exceeding 0.9, they typically require continuous monitoring of complex physiological parameters, resulting in high implementation costs that may not align with the practical conditions of most healthcare institutions (Hoque et al., 2024). In contrast, our model does not rely on expensive continuous monitoring equipment, making it more suitable as an early screening tool. This design feature not only significantly lowers the barrier to implementation but also allows for seamless integration into routine clinical workflows across multiple departments.

Innovation

The innovation of this study lies in several aspects. First, it focused on symptomatic UTIs and used a multifactorial analysis to systematically explore the effects of nine factors on it, addressing the limitations of previous single-factor studies. Second, the study applied the latest Delphi consensus criteria for diagnosing symptomatic UTIs, improving diagnostic accuracy and scientific validity. Additionally, the study uniquely analyzed differences in infection rates between departments, revealing the impact of hospital cross-infection and antibiotic use habits on symptomatic UTIs, providing important insights for hospital infection management.

Limitations

As a retrospective study, this research is subject to certain limitations, including potential selection and information biases. The sample size is relatively limited, especially among female patients. Moreover, the diagnosis of symptomatic UTIs was based mainly on clinical and laboratory criteria, which may have resulted in incomplete identification of some infection cases. Future studies should examine the differential effects of long-term indwelling catheter use versus intermittent catheterization on symptomatic UTI incidence and investigate optimal catheterization strategies. It is also important to note that urodynamic data, which are valuable for evaluating urinary tract infections, were not consistently documented in the admission records for this project.

Conclusions

This study identified prolonged hospital stay and cumulative antibiotic use as independent predictors of symptomatic UTIs in traumatic SCI patients. Risk factors for symptomatic UTIs include indwelling catheterization, admitting department, polytrauma, and AIS grade. The predictive model for symptomatic UTIs based on these risks demonstrated strong discriminatory capacity and clinical interpretability, with an AUC of 0.81, outperforming existing tools that rely on specialized assessments or expensive biomarkers. Key implementation strategies involve integrating automated risk alerts into electronic health records to guide antibiotic stewardship and targeted catheterization protocols, especially in high-risk departments. These findings advocate for a shift from reactive treatment to preventive care, focusing on reducing hospital stays and enhancing multidisciplinary antimicrobial oversight. This approach supports global efforts to combat antimicrobial resistance while improving outcomes for TSCI patients.

Supplemental Information

Supplemental Information 1 Raw code of ROC analysis and curves drawing

Supplemental Information 2 The detailed process of analysis in R

Supplemental Information 3 Dataset of ROC analysis and curves drawing

Supplemental Information 4 Results of VIF and Bootstrap tests

The authors acknowledge the contribution of all the participants and collaborators of this study.

Additional Information and Declarations

Competing Interests

Author Contributions

Human Ethics

Data Availability

The authors declare there are no competing interests.

Huayong Du conceived and designed the experiments, performed the experiments, analyzed the data, prepared figures and/or tables, authored or reviewed drafts of the article, and approved the final draft.

Zehui Li conceived and designed the experiments, performed the experiments, analyzed the data, prepared figures and/or tables, authored or reviewed drafts of the article, and approved the final draft.

Jinming Zhang conceived and designed the experiments, performed the experiments, analyzed the data, prepared figures and/or tables, authored or reviewed drafts of the article, and approved the final draft.

Xiaoxin Wang conceived and designed the experiments, performed the experiments, prepared figures and/or tables, authored or reviewed drafts of the article, and approved the final draft.

Yingli Jing conceived and designed the experiments, performed the experiments, analyzed the data, prepared figures and/or tables, authored or reviewed drafts of the article, and approved the final draft.

Degang Yang conceived and designed the experiments, performed the experiments, analyzed the data, prepared figures and/or tables, authored or reviewed drafts of the article, and approved the final draft.

Jianjun Li conceived and designed the experiments, performed the experiments, analyzed the data, prepared figures and/or tables, authored or reviewed drafts of the article, and approved the final draft.

The following information was supplied relating to ethical approvals (i.e., approving body and any reference numbers):

The China Rehabilitation Research Center granted Ethical approval to carry out the study within its facilities (Approval No. 2022-143-01).

The following information was supplied regarding data availability:

The raw measurements are available in the Supplemental File.

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
