# Peer review of "Traumatic spinal cord injury: identifying independent risk factors and predictive model development for symptomatic urinary tract infections"

_PeerJ, doi:10.7717/peerj.19473_

## Round 0.1 · original submission · Major Revisions

You must respond in detail to the comments from all three reviewers.

·

Basic reporting

The authors have conducted a study on spinal cord injury patients to identify the risk factors for developing urinary tract infections in them.
1. There is unnecessary capitalisation of letters for some words. Eg: Symptomatic (lines 11, 15, etc.)
2. Line 126 - no symptomatic UTI may be rephrased as asymptomatic UTI

Experimental design

1. I understand that the article forms a part of the big project - A Cohort Study on the Predictive Role of Clinical Factors and the Urethral Microbiome in Recurrent Urinary Tract Infections in Hospitalized Rehabilitation Patients with Spinal Cord Injury (lines 40-42). Have the findings from this project been already published?
2. It is not clear whether this is a retrospective or prospective study (45).

Validity of the findings

1. Please explain how you calculated the risk score (168-169).
Risk Score=0.014× Antibiotic uses+0.006×Length of stay+0.249×Admitting department20.234× Injury level
How did you incorporate the text variables in the score?

2. The study uses multifactorial analysis and Delphi consensus criteria for diagnosis, enhancing validity

Additional comments

No comment

Reviewer 2 ·

Basic reporting

The authors studied to identify independent risk factors and develop a predictive model for symptomatic UTIs in TSCI patients. This study focused on very important matters to manage symptomatic UTIs in TSCI patients.

Experimental design

1) In this institute, are the long-term indwelling catheters replaced before initiating antimicrobial therapy for symptomatic urinary tract infection?
2) As risk factors to analyze, renal function and Performance Status (E.R., KPS) should be included.

Validity of the findings

Discussion: It should be described in more detail about prevention and management strategies of symptomatic UTIs in TSCI patients.

Additional comments

None

Reviewer 3 ·

Basic reporting

No comment

Experimental design

Originality

This study advances the field by developing a novel risk score model (AUC = 0.80) for early UTI prediction. Furthermore, it uses a Delphi consensus criteria for symptomatic UTI, which enhances diagnostic reliability.
As a minor observation, the manuscript does not explicitly compare its risk score model to existing UTI risk assessment tools; therefore, the authors should discuss how the new model compares with other risk scoring systems regarding clinical applicability and performance.


Research Question

The research aim and question are clear, objective, and clinically relevant.
However, the manuscript does not emphasize how the findings could influence treatment decisions or infection prevention protocols (risk score interpretation);

Experimental design

Ethical standards are fulfilled and comply with the journal and scientific requirements.


If available, comorbidities and lifestyle factors (renal dysfunction, mobility levels, hydration ( ingesta/excreta)) could be considered in the analysis, which can influence UTI in SCI.

The statistical analysis is well performed using univariate and multivariate logistic regression; however, a multicollinearity assessment should be performed to determine each predictor's actual effect and exclude bias in regression coefficients and incorrect conclusions about the risk factors ( the most common method is variance inflation factor). Furthermore, it could provide insights regarding redundant predictors and improve prediction accuracy and reproducibility.

The predictive model’s performance is assessed using AUC (0.80), but no internal validation for example, cross-validation, bootstrap resampling) is reported. The authors should consider this approach to confirm the model's stability.

Furthermore, the authors should mention if they used the analysis of the goodness-of-fit tests to evaluate model reliability.

Replicability

The study provides enough details for replicability.

The catheterization methods/protocol description is missing and must be added.

Validity of the findings

Impact, Novelty & Meaningful Replication

The manuscript describes data sources, inclusion criteria, and statistical methods in sufficient detail to make future replication feasible. It also presents a new predictive model for symptomatic UTIs in SCI patients, integrating multiple independent risk factors.

Data Robustness

The statistical analysis was well performed; however, it could have been improved as stated before.

Conclusions, Relevance, limitations

Based on their findings, the authors highlight UTI prevention strategies (e.g., reducing unnecessary antibiotic use and optimizing hospital stay durations).

However, while the study provides a valuable risk score model, the conclusion does not suggest how clinicians can integrate it into practice.

---

## Round 0.2 · Minor Revisions

Please revise your manuscript according to the reviewer's comments.
Yours,
Yoshi
Prof. Yoshinori Marunaka, M.D., Ph.D.

·

Basic reporting

Thank you for the corrections.

The term "no symptomatic UTI" is grammatically incorrect and is not acceptable on its own without a proper explanation.
Please include a sentence in the article explaining what you mean. Eg: Patients classified as 'no symptomatic UTI' included both those without any urinary tract infection and those with asymptomatic bacteriuria.
Once that explanation is given, you can continue to use the term "no symptomatic UTI."

Experimental design

No comment

Validity of the findings

No comment

---

## Round 0.3 · accepted · Accept

Congratulations.

Yours,
Yoshi
Prof. Yoshinori Marunaka, M.D., Ph.D.

·

Basic reporting

I thank the authors for making the necessary modifications suggested earlier.

Experimental design

No comment.

Validity of the findings

No comment.

Additional comments

No comment.